# Understanding Neuropathy Features in the Context of Nitrous Oxide Abuse: A Combined Electrophysiological and Metabolic Approach

**DOI:** 10.3390/biomedicines12020429

**Published:** 2024-02-14

**Authors:** Guillaume Grzych, Marie Scuccimarra, Laura Plasse, Emeline Gernez, Francois Cassim, Benjamin Touze, Marie Girot, Cécile Bossaert, Céline Tard

**Affiliations:** 1Biology Center, Biochemistry Department, Lille University Hospital, 59000 Lille, France; laura.plasse@chu-lille.fr (L.P.); emeline.gernez@chu-lille.fr (E.G.); ben.touze57@gmail.com (B.T.); 2IMPact de l’Environnement Chimique sur la Santé (IMPECS), University of Lille, 59000 Lille, France; 3Reference Center for Neuromuscular Diseases, Neurology Department, Lille University Hospital, 59000 Lille, France; mscuccimarra@for.paris (M.S.); celine.tard@chu-lille.fr (C.T.); 4Neurophysiology Department, Lille University Hospital, 59000 Lille, France; francois.cassim@chu-lille.fr; 5Emergency Department, Lille University Hospital, 59000 Lille, France; marie.destee@chu-lille.fr (M.G.); cecile.bossaert@chu-lille.fr (C.B.)

**Keywords:** nitrous oxide, neurological damage, metabolism, cobalamin, homocysteine, methylmalonic acid, neurology

## Abstract

Background: The incidence of neurological complications associated with nitrous oxide (N_2_O) abuse, including N_2_O-induced myelopathy and neuropathy, has risen in the past decade. N_2_O-induced neuropathy often presents as a subacute axonal pathology; however, demyelinating patterns mimicking Guillain–Barré syndrome have also been observed. This study explores the metabolic pathophysiology of N_2_O-induced neuropathy, focusing on the alteration in metabolism to provide a deeper understanding of the biochemical pathways influencing the diverse electrophysiological patterns observed. Methods: We conducted a combined metabolic and electrophysiological exploration of 35 patients who underwent electromyographic exams at our referral center over a three-year period for sensorimotor symptoms linked to recreational N_2_O use. We collected demographic, clinical, radiological, electrophysiological, and biological data. Patients were categorized into axonal or demyelinating groups based on their electrophysiological patterns, and metabolic parameters were compared. Results: Our cohort predominantly exhibited a length-dependent sensorimotor axonal symmetrical neuropathy affecting the lower limbs. Among the patients, 40% met the demyelinating criteria, with four patients exhibiting conduction blocks. The demyelinating group had a significantly higher peripheral neuropathy disability (PND) score at diagnosis. Elevated homocysteine and methylmalonic acid (MMA) levels were noted in all patients, but these were lower in the demyelinating group. Conclusions: This study highlights the diverse electrophysiological manifestations of N_2_O-induced neuropathy and underscores the potential role of metabolic parameters as biomarkers to understand its pathophysiology. Lower hyperhomocysteinemia and MMA levels were observed in demyelinating patterns. In this study, we did not observe further improvement, but it is well-known that demyelinating features have a better prognosis related to the further remyelination. These findings contribute to a better understanding of N_2_O-related neuropathic damage and could guide future therapeutic interventions based on biochemical–neurophysiological stratifications.

## 1. Introduction

Nitrous oxide (N_2_O) abuse induces peripheral neurological damage through a spectrum of clinical presentations, ranging from mild sensory disturbances to severe motor deficits. Recent advances in our understanding of these conditions have elucidated the critical role of cobalamin (vitamin B12) metabolism in maintaining the integrity of the peripheral nervous system. Cobalamin is a necessary cofactor for key enzymatic reactions within the nervous tissue. Deficiencies in cobalamin, whether due to dietary insufficiency, malabsorption, or interference by environmental toxins, such as nitrous oxide, can disrupt these pathways. Nitrous oxide (N_2_O), a colorless and non-flammable gas, is extensively utilized in medical settings for its established analgesic, anesthetic, and anxiolytic properties. [1]. Commonly referred to as “laughing gas” or “whippits”, nitrous oxide is recognized for its transient recreational effects, including euphoria, altered perception, and a sense of dissociation, which typically subside within a few minutes. [2]. However, the misuse of this substance has escalated significantly in recent years, positioning it as one of the most commonly abused psychoactive agents among young individuals across Europe [3]. 

The pervasive recreational utilization of nitrous oxide and the increase in consumption rates, both quantitatively and frequency-wise, have led to an uptick in chronic toxicity cases in recent years. There has been an exponential rise in hospitalizations due to neurological complications induced by N_2_O, such as acute or subacute tetraparesis, which are often attributed to N_2_O-induced neuropathy and/or myelopathy. N_2_O-induced neuropathy typically presents as a length-dependent sensorimotor axonal neuropathy characterized by profound motor axonal loss predominantly affecting the lower limb [4,5,6,7]. Furthermore, subacute demyelinating patterns have been observed, demonstrating clinical and electrophysiological features that may occasionally mimic those of Guillain–Barré syndrome, also called acute inflammatory demyelinating polyradiculoneuropathy (AIDP) [8,9]. The distinction is important not only for diagnosis but also for prognosis. We know that demyelinating lesions often have a better prognosis given the potential for remyelination (especially if the causal aspect is stopped) while axonal loss is permanent. In the case of axonal loss, motor recovery is only possible by terminal reinnervation (and sensory recovery rather than by neuroplasticity), mechanisms slower and more limited than remyelination.

Thus, axonal damage is more severe. However, the initial clinical severity assessed by clinical examination scores does not make it possible to distinguish between these two pathophysiological processes. Only electroneuromyography allows this distinction. In primary demyelinating disorders, we also know that there can be secondary damage, and when the axonal damage is too severe, the signs of demyelination can no longer be evaluated. Thus, demyelinating damage could also occur earlier in N_2_O-induced neuropathies.

The pathophysiology of N_2_O-induced toxicity is linked to the functional inhibition of vitamin B12, also known as cobalamin (Figure 1). Nitrous oxide acts as a potent oxidizing agent, leading to the oxidation of the cobalt ion (Co+) within cobalamin(I), converting it to the Co3+ state and subsequently generating cobalamin (II), which is incapable of accepting methyl groups [10]. This leads to a decrease in available methylcobalamin, a necessary cofactor for the enzyme methionine synthase. As a result, there is a subsequent decrease in the activity of methionine synthase, which plays a pivotal role in catalyzing the transformation of homocysteine into methionine [11]. The authors propose that the accrual of cobalamin (II), which also acts as a cofactor for methionine synthase reductase (MTRR), promotes the enhancement of MTRR activity [12]. This enzyme-mediated process facilitates the conversion of cobalamin (II) to cobalamin (III) via S-adenosylmethionine, thereby increasing the production of homocysteine [1]. 

Related to the two previously described mechanisms, patients who have consumed N_2_O typically exhibit markedly elevated levels of homocysteine. Plasma homocysteine levels are highly sensitive and serve as an indicator of recent N_2_O exposure, given their rapid increase following consumption [13]. Nonetheless, this biomarker is not exclusive to N_2_O intoxication; elevated levels may also occur in instances of vitamin B9 or B12 deficiencies, renal insufficiency, or hereditary metabolic disorders.

The impact of nitrous oxide (N_2_O) on the activity of methylmalonyl-CoA mutase, an enzyme facilitating the conversion of methylmalonic acid (MMA) to succinyl-CoA, remains a subject of ongoing debate. It has been hypothesized that the oxidation of cobalamin(I) by N_2_O may induce a generalized cobalamin deficit, encompassing a reduction in adenosylcobalamin—a cofactor essential for methylmalonyl-CoA mutase function. Although elevations in MMA levels are not consistently observed and, thus, do not serve as a reliable marker for N_2_O misuse, they are more specifically indicative of disturbances in vitamin B12-dependent pathways. Moreover, there appears to be a correlation between increased MMA levels and the clinical severity of N_2_O intoxication [13].

While the characteristics of neurological complications associated with nitrous oxide (N_2_O) exposure are extensively documented, the relationship between distinct peripheral neurological patterns and their underlying pathophysiological processes remains to be elucidated. This study aimed to improve understanding of N_2_O pathophysiology by examining metabolic alteration in patients presenting with axonal versus demyelinating patterns of N_2_O-induced neuropathy and myelopathy. By providing an in-depth analysis of the biochemical pathways that contribute to the observed electrophysiological variations, this research could potentially inform the development of targeted therapeutic strategies.

## 2. Materials and Methods

### 2.1. Patient Selection and Data Collection 

Patients with N_2_O abuse (last declared N_2_O consumption is less than 1 week ago) treated in the CHU de Lille (from the Emergency Unit or general practice consultations) between March 2020 and April 2023, who underwent electromyography in our referral center, were included in the study. The study protocol was assigned the declaration number DEC21-356. In accordance with the French National Data Protection Commission regulation, written informed consent was not required for this non-interventional retrospective study. The data collected were extracted from hospital medical records. The exclusion criteria were as follows: patients with inherited metabolic disorders, renal insufficiency, folate deficiency, total vitamin B12 > 5 ng/mL (possible self-medication), and last declared N_2_O consumption more than 1 week ago.

### 2.2. Clinical Data

Clinical features collected included the presence of sensory symptoms, motor weakness (if present, affecting the lower and/or upper limbs, distal and/or proximal), gait abnormality, and deep tendon reflexes (classified as normal, absent for Achilles reflexes, absent in the lower limbs, or absent in all four limbs). The patients were then classified according to their clinical phenotypes: length-dependent vs. non-length-dependent, sensory or sensorimotor, and the functional evaluation carried out with the peripheral neuropathy disability (PND) (ranging from I, indicating sensory disturbances with preserved walking capability, to IV, representing confinement to a wheelchair or being bedridden, Table 1). Concerning neurophysiological data, the criteria of Hadden et al. were used to define patients with demyelinating criteria [14]. As usual, patients with no demyelinating criteria were classified as “axonal” concerning the group of neuropathy. However, as stated in the introduction, it is possible for a patient with a demyelinating pattern to also have an axonal loss, defined as a decrease in amplitude in the nerve conduction studies (defining sensory axonal loss and/or motor axonal loss).

### 2.3. Biological Measurements

Concurrent with clinical evaluation, quantitative determinations of vitamin B12, B9 (folate), homocysteine, methylmalonic acid (MMA), and methionine were conducted for each patient to ensure temporal consistency between biochemical and clinical assessments. For homocysteine and MMA, blood samples were collected in EDTA tubes. Plasma was aliquoted and then frozen and stored at −20 °C until analysis. Homocysteine was measured by high-performance liquid chromatography on UFLC XR (Shimadzu, Kyoto, Japan) associated with tandem mass spectrometry using MRM on API 3200 Q TRAP (AB SCIEX, Framingham, MA, USA). MMA was measured by ultra-performance liquid chromatography associated with tandem mass spectrometry, using UPLC I-CLASS/XEVO TQXS1 (Waters Corp., Milford, MA, USA). Plasma homocysteine increase is defined as >15 µmol/L and plasma MMA increase is defined as >0.4 µmol/L.

Vitamin B9 and B12 were measured using blood samples collected in a clot activator tube (CAT). Their concentration was determined by chemiluminescent immunoassay on a UNICEL DXI 800 (Beckman Coulter Inc., Brea, CA, USA). Serum total vitamin B12 deficiency was defined as <0.2 ng/mL, and serum folate deficiency was defined as <3.1 ng/mL.

### 2.4. Electrophysiological Data (EDX)

Electrophysiological assessments were systematically carried out by a certified neurophysiologist for all participants, utilizing the NATUS (Middleton, WI, USA) Dantec Keypoint apparatus (center of reference of neuromuscular disorders). A comprehensive sensory and motor nerve conduction study was performed, including the fibular and median nerves for motor conduction, while the sural and median nerves were assessed for sensory conduction. The inclusion of additional nerve conduction studies was at the discretion of the examining neurophysiologist. 

For each participant, electromyographic examinations were reviewed to ascertain compliance with Hadden’s criteria for acute inflammatory demyelinating polyneuropathy (AIDP) [14], as well as for axonal and demyelinating neuropathies in accordance with the classification system proposed by Albers [15]. According to Albers, a reduced distal motor amplitude <80% of the norm on at least 2 nerves defines an axonal pattern, whereas a prolonged distal motor latency >120% of the upper limit of the norm, and/or proximal/distal motor amplitude ratio and/or motor conduction velocity < 80% of the norm, on at least 2 nerves, defines a demyelinating pattern. F-wave latency was not used in the classification for demyelinating ENMG because it was not systematically studied.

AIDP criteria were used to determine the groups as “demyelinating” and “axonal” for further comparisons.

### 2.5. Statistical Analyses

Statistical analyses were performed using JASP Team (2020) (JASP, Amsterdam, The Netherlands) and JASP (Version 0.18.1) (JASP. Group comparisons were performed using the independent samples *t*-test for normally distributed data and the Mann–Whitney U test for non-normally distributed data. Statistical significance was defined at a *p*-value threshold of <0.05.

## 3. Results

### 3.1. Selected Patients Exhibit a Spectrum of Clinical Severities, with a Predominance of Gait Disturbances

Within the biological database of 113 patients, 35 fulfilled the inclusion criteria. Of these, 29 patients, accounting for 82.9%, were male. The median age at the time of diagnosis was 21 years (interquartile range: 17–29 years) (Table 2).

Clinical characteristics of the cohort are concisely presented in Table 2. Notably, 11 patients (31.4%) exhibited exclusively sensory symptoms, while 24 experienced combined sensorimotor deficits. Pure motor symptoms were absent in this group. A large majority of the patients (88.6%) demonstrated an abnormal gait (Table 2).

### 3.2. Selected Patients Show Metabolic Alterations Characteristic of Nitrous Oxide Abuse

Biological parameters, as detailed in Table 3, underscore the metabolic impact of nitrous oxide exposure within the study cohort. Consistently elevated homocysteine levels were observed across all patients, with a mean concentration of 104 µmol/L. This elevation in homocysteine level is concomitant with increased levels of methylmalonic acid (MMA), further corroborating the disruption of normal cobalamin metabolism typically associated with nitrous oxide intoxication. Concurrently, vitamin B12 levels were found to be reduced, affirming the anticipated inverse relationship between B12 concentrations and homocysteine/MMA levels. Notably, folate (vitamin B9) and methionine levels remained within normal parameters, suggesting a specific interference with the cobalamin-dependent pathways rather than a generalized disruption of one-carbon metabolism. These biochemical findings provide a nuanced insight into the metabolic derangements precipitated by nitrous oxide abuse, underpinning the clinical manifestations observed in the patient population.

### 3.3. Most of Included Patients Exhibited Myelopathy on MRI and Axonal Loss on Electromyography

Medullary MRI was performed on 33 patients (94.3%), revealing myelopathy in 23 (69.7%) of cases. 

Electrophysiological findings (Table 4) show that three patients (8.6%) had a normal EDX. 

Thirty-four percent of patients met the criteria of AIDP. Fourteen patients had demyelinating features (40%), thirteen had an axonal loss (37%), and one patient exhibited isolated demyelinating neuropathy without axonal involvement. Then, an “axonal” pattern was defined for 21 patients (60%). 

An axonal loss was observed in the vast majority of patients (Table 4): 89% of patients had an axonal loss, motor loss occurred in 80% of patients, sensory loss occurred in 37% of patients, and sensorimotor loss occurred in only 1 patient (3% of patients). Active denervation was observed in 35% of patients.

The EDX injuries were mainly exclusively motor (16 patients, 46%), sensorimotor (12 patients, 34%), and rarely pure sensory (in one patient, 3%). 

### 3.4. Our Study Finds no Demographic Differences, Higher PND Scores and Lower Homocysteine in the Demyelinating Group between the Axonal and Demyelinating Groups

In the present study, a comprehensive analysis was conducted to compare demographic and clinical parameters across the study groups. Notably, our analysis revealed no significant differences in age and gender distribution among these groups. This demographic parity is crucial as it mitigates potential biases related to these variables in the interpretation of our findings. 

Further investigation into clinical severity, as assessed by the PND score, indicated a markedly higher level of clinical severity in the demyelinating group. The statistical significance of this observation was confirmed with a Mann–Whitney U test result of U = 89.5 and a *p*-value of 0.028. This finding suggests a more pronounced clinical manifestation in the demyelinating group compared to their counterparts (Table 5).

Additionally, our study delved into the comparison of metabolic markers across the groups. A significant finding was the lower level of homocysteine in plasma in the demyelinating group, (*p*-value = 0.045). This outcome highlights a potentially noteworthy biochemical difference between the groups. 

However, when examining other metabolic markers, including methionine, methylmalonic acid (MMA), and vitamin B12, the analysis did not reveal any statistically significant differences between the groups. These results are detailed in Table 5. The absence of significant disparities in these markers suggests that the noted differences in clinical severity and homocysteine levels are not mirrored across all examined biochemical parameters.

## 4. Discussion

The primary objective of this study was to enhance our understanding of the pathophysiological mechanisms underlying nitrous oxide (N_2_O) exposure and its effects on peripheral neurology. Our research specifically aimed to delineate the relationship between distinct neurological patterns induced by N_2_O exposure and their biochemical correlates. We focused on patients presenting with axonal versus demyelinating patterns of N_2_O-induced neuropathy and myelopathy, examining metabolic changes and their impact on the peripheral nervous system. A critical aspect of this study was the investigation of cobalamin (vitamin B12) metabolism, essential for the integrity of the nervous system and its disruption due to N_2_O exposure. This included analyzing elevated plasma homocysteine levels, indicative of recent N_2_O exposure, and the effect of N_2_O on methylmalonyl-CoA mutase activity.

The study aimed to provide a detailed analysis of the clinical and electrophysiological features of N_2_O-induced neurological damage, with a focus on understanding the biochemical pathways that contribute to these conditions. By exploring these pathways, the research sought to inform the development of targeted therapeutic strategies for N_2_O-induced neuropathy and myelopathy. This endeavor is particularly critical given the increasing prevalence of N_2_O misuse and the need for a deeper understanding of the pathophysiological basis for the varied neurological manifestations associated with its abuse.

From a clinical perspective, similar to other series in the literature, we primarily observe a sensorimotor axonal electromyographic profile with a motor or predominantly motor pattern that is length-dependent. Demyelination signs are found in 34–40% of patients, according to our criteria, most often presenting as mixed axonal–myelinic involvement (secondary axonal loss). In our cohort, patients with demyelinating features exhibit significantly higher PND scores. These demyelinating mechanisms may play a role in more subacute neuropathy, characterized by rapidly evolving symptoms and leading to greater functional impairment at diagnosis. However, a better prognosis may be expected in cases with higher motor amplitude in the common fibular and tibial nerves. A previous published study indicated a correlation between longer exposure duration and more prolonged symptoms with lower motor amplitudes. Further follow-up studies of these patients are needed, as there is a keen interest in understanding their outcomes in subgroups [7].

The demyelinating abnormalities observed in EMG are essentially motor-based. This is linked to a methodological pitfall because the criteria for demyelination are more debated for sensory nerves. We should propose more a more systematic study of sensory evoked potentials for ataxic patients with normal sensory amplitudes on the peripheral nerves to better study on the sensory radicular damage of these patients and to search for proximal demyelination (as for AIDP).

Concerning the patients with normal EDX, we can suppose that (i) the EDX was too early, (ii) the patient only had central damage, or (iii) the patient only had a small fiber neuropathy (not evaluated by EDX study). On the other hand, we have to highlight the frequency of axonal loss (89% of patients, 80% with motor one), explaining the poor prognosis of this neuropathy. 

The biological characteristics of the patients included in the study present a pattern indicative of nitrous oxide-related metabolic disturbance. Elevated homocysteine and methylmalonic acid (MMA) levels were a consistent finding, reflecting the direct impact of nitrous oxide on the metabolism of cobalamins. Specifically, homocysteine concentrations were significantly above the normal range, with a mean level of 104 µmol/L, highlighting a substantial deviation from standard values. Meanwhile, the observed decrease in vitamin B12 levels among these patients aligns with the known effects of nitrous oxide on vitamin B12 inactivation and subsequent functional deficiency. Contrary to the alterations seen in homocysteine and MMA, the levels of folate (vitamin B9) and methionine remained within the normal range across the cohort. This suggests that the metabolic perturbations are specific to the pathways involving vitamin B12, rather than a broad impairment of the one-carbon metabolic network. Collectively, these biological characteristics provide valuable insights into the biochemical sequelae of nitrous oxide exposure and underscore the need for targeted metabolic evaluations in affected individuals.

No differences were also found between the levels of MMA (methylmalonic acid) and methionine, which may seem contradictory to certain previously published studies where MMA levels increased and methionine levels decreased with clinical severity [13,16,17]. However, in these studies, general clinical involvement was assessed without sub-categorization according to types of neurological involvement. These results suggest that metabolic alterations in the monocarbon pathway do not condition the type of neurological involvement, necessitating the investigation of still unknown pathways, such as alterations in antioxidant defenses. Indeed, nitrous oxide is a potent oxidizing agent that can lead to significant oxidative stress, potentially causing peripheral neurological damage [18]. The only metabolic difference found between the axonal and demyelinating groups is a lower homocysteine level in demyelinating patients. However, the role of homocysteine in the direct neurological toxicity of nitrous oxide intoxication remains unclear. The fact that homocysteine is found to be higher in axonal involvement, which has a poorer prognosis, could suggest its toxicity to certain types of neuronal cells. Nevertheless, the kinetics of homocysteine in intoxication are still poorly understood and appear to be more directly related to the kinetics of nitrous oxide rather than clinical severity [17]. Kinetic studies on these markers are, therefore, necessary to better understand their implications.

The group with the demyelination criterion had a more severe clinical presentation on admission than the pure axonal group with nevertheless lower hyperhomocysteinemia. This is in good agreement with the fact that hyperhomocysteinemia reflects secondary axonal damage. This goes with the hypothesis of a primary demyelinating neuropathy: thus, patients who have a subacute attack consult earlier, as they are faced with more serious signs, than those who have a chronic attack, more progress more slowly but with a much worse prognosis because the axonal lesions are more severe. We could assume here that the subacute demyelinating pattern differences are the result of consumption (i.e more intense/impulsive consumption with rapid initial demyelination) or perhaps that there are associated inflammatory factors.

Our study has several limitations. Since our cohort consists solely of patients who underwent electromyography at our referral center, we predominantly report cases with severe N_2_O-induced neurological pathologies (100% were hospitalized patients, 89% had gait disorders, and 30% had a PND score greater than IIIa). This may represent only the tip of the iceberg: a large survey involving over 240,000 individuals found that 17% of participants reported chronic use of N_2_O, with 4.2% of recreational users experiencing persistent paresthesia or numbness in a stocking-glove pattern, indicative of a length-dependent axonal polyneuropathy [16]. Investigating metabolic disorders in these patients with milder symptoms could be highly informative. Similar to most studies on this topic, our study is limited by a small patient cohort and a lack of long-term follow-up, attributed to the challenges in tracking patients who often refuse care.

This study provides preliminary data that underscore the necessity for further confirmation through research on a larger patient cohort. Additionally, our findings suggest the potential need for exploring the neurotoxic effects of homocysteine. The limited scale of our current investigation highlights the importance of expanding our research scope to not only validate these initial observations but also to delve deeper into understanding the intricate relationship between elevated homocysteine levels and the possible neurotoxic implications. Future studies with expanded cohorts are crucial for establishing definitive conclusions and enhancing our comprehension of homocysteine role in nitrous oxide abuse.

## 5. Conclusions

Our study contributes to the understanding of the neuropathological mechanisms associated with nitrous oxide (N_2_O) intoxication. We have highlighted some intricate relationship between cobalamin metabolism and the development of both demyelinating and axonal injuries. Our findings underscore the critical role of cobalamin in maintaining neural integrity and how its disruption due to N2O exposure can lead to substantial neurological impairment.

## Figures and Tables

**Figure 1 biomedicines-12-00429-f001:**
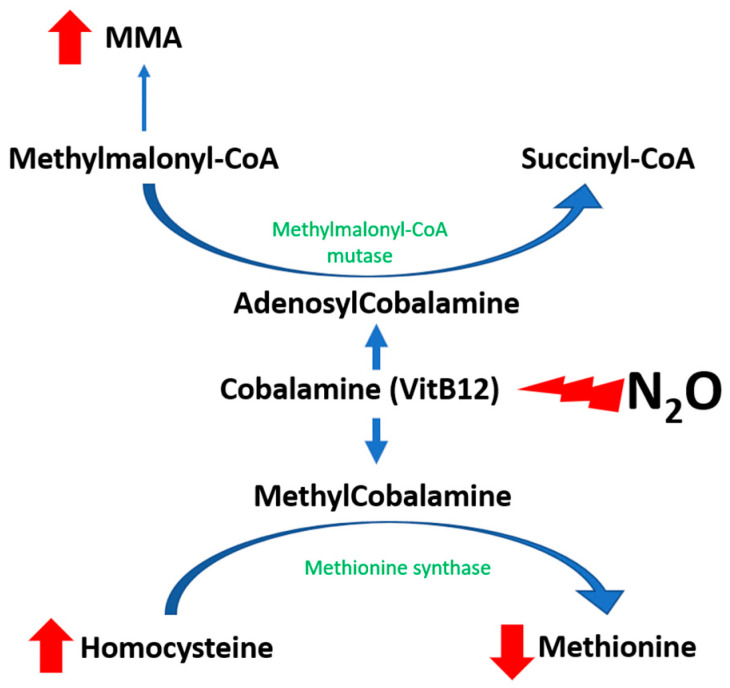
Hypothetical impact of N_2_O on metabolism. Red upward arrow: increase; red downward arrow: decrease, red thunder symbol: oxidation.

**Table 1 biomedicines-12-00429-t001:** Peripheral neuropathy disability (PND) score. PND score according to clinical symptoms.

PND Score	Symptoms
0	Asymptomatic
I	Sensory symptoms, walking normally
II	Walking difficulties, does not require support or stick
IIIa	One stick or crutch needed for ambulation
IIIb	Two sticks or two crutches needed for ambulation
IV	Patient confined to a bed or wheelchair

**Table 2 biomedicines-12-00429-t002:** Clinical and neurological characteristics of patients included in the study.

Neurological Characteristics	*n*/*n* Tested (%)
**Motor examination**	
Distal decreased muscle strength in lower limbs	22/35 (62.8)
Proximal decreased muscle strength in lower limbs	12/35 (34.3)
Distal decreased muscle strength in upper limbs	10/35 (28.6)
Proximal decreased muscle strength in lower limbs	6/35 (17.1)
**Sensory examination**	
Paresthesia/hypoesthesia in lower limbs	35/35 (100)
Romberg sign: ataxia	29/30 (96.7)
**Deep tendon reflexes (DTR)**	
Absent Achilles DTR	27/33 (81.8)
Absent in the lower limbs	23/33 (69.7)
Absent in the upper limbs	9/33 (27.3)
**Gait disorders**	31/35 (88.6)
Length-dependent symptoms	18/34 (52.9)
**PND Score (*n*/*n* tested, %)**	
PND I	3/35 (8.6)
PND II	21/35 (60)
PND IIIa	2/35 (5.7)
PND IIIb	7/35 (20)
PND IV	2/35 (5.7)

**Table 3 biomedicines-12-00429-t003:** Biological characteristics of patients included in the study. MMA: methylmalonic acid; Min: minimum; max: maximum.

Parameters	Mean	Standard Deviation	Min	Max	Reference Range
Vitamin B12	0.160	0.137	0.05	0.72	>0.2 ng/mL
Vitamin B9	8.6	4.1	3.3	22.3	>3.1 ng/mL
Homocysteine	104.7	50	25	250	<15 µmol/L
Methionine	18.75	5.8	4	36	16–29 µmol/L
MMA	6.38	6.06	0.24	19.87	<0.4 µmol/L

**Table 4 biomedicines-12-00429-t004:** Electrophysiological data of selected patients.

Electrophysiological Data	*n*/*n* Tested (%)
**Demyelinating criteria (Albers)**	14/35 (40)
**AIDP criteria (Hadden)**	12/35 (34)
**Motor nerve conduction**	
Reduced motor amplitudes	28/35 (80)
Slowing of motor conduction velocities	25/35 (71.4)
Conduction bloc	4/35 (11.4)
Prolonged motor distal latencies	9/35 (25.7)
**Sensory nerve conduction**	
Reduced sensory amplitude	13/35 (37.1)
**Needle electromyography**	
Signs of active denervation (fibrillation or positive slow waves)	12/34 (35.3)

**Table 5 biomedicines-12-00429-t005:** Comparative analysis of demographic and metabolic parameters between axonal and demyelinating groups. Data are expressed as mean ± standard deviation (SD). *p*-value were obtained by non-parametric Mann–Whitney U test. PND: peripheral neuropathy disability; B12: vitamin B12; MMA: methylmalonic acid; HCY: homocysteine.

Parameters	Axonal Group	Demyelinated Group	*p*-Value
Gender (M/F)	17/6	10/2	0.34
Age (years)	21.5 (2.6)	22.3 (3.1)	0.18
PND score	2.2 (0.9)	3.0 (1.0)	0.028
B12 (ng/mL)	0.135 (0.075)	0.223 (0.226)	0.98
MMA (µmol/L)	6.67 (6.67)	6.01 (5.51)	0.32
Methionine (µmol/L)	17.68 (5.98)	21.00 (5.02)	0.34
HCY (µmol/L)	114+/−46	90+/−53	0.04

## Data Availability

On requirement.

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
