# Peer review of "Understanding Neuropathy Features in the Context of Nitrous Oxide Abuse: A Combined Electrophysiological and Metabolic Approach"

_biomedicines, 2024, doi:10.3390/biomedicines12020429_

Round 1

Reviewer 1 Report

Comments and Suggestions for Authors

It is a well written paper on interesting subject of N2O abuse. The introduction covers all necessary information so that the reader can have a full understanding of the problem. Methods are well described and summarized. Discussion is broad with good insight into the problem. I believe that this paper is ready for publishing after minor spell check as there are couple of spelling errors in the text.

Comments on the Quality of English Language

 Minor editing of English language required

Author Response

Thank you for your thoughtful and constructive review. We appreciate your positive feedback on the comprehensive nature of our introduction, the clarity of our methods section, and the depth of our discussion on the subject of N2O abuse. Your comments affirm our commitment to contributing valuable insights to the literature. We will conduct a thorough spell check to address and correct the spelling errors identified. We are grateful for your recommendation for publishing and will make the necessary minor corrections promptly.

Reviewer 2 Report

Comments and Suggestions for Authors

This study significantly contributes to our understanding of the metabolic pathophysiology related to nitrous oxide (N2O)-induced neuropathy. The increasing occurrence of neurological complications associated with N2O abuse, particularly myelopathy and neuropathy, highlights the importance of this research.

The strength of this study lies in its combined metabolic and electrophysiological approach, which comprehensively analyzes the biochemical and electrophysiological changes related to N2O-induced neuropathy. The utilization of a clearly defined patient cohort, coupled with detailed data collection and categorization based on electrophysiological patterns, enhances the reliability of the findings.

However, there are several limitations and weaknesses that require attention. Firstly, the relatively small sample size of patients may limit the generalizability of the results. It is advisable for the authors to provide justification and methodology for the sample size calculation. Secondly, the absence of a control group in this study poses a challenge in specifically attributing the observed metabolic and electrophysiological changes to N2O exposure. Future research should consider incorporating a control group without N2O exposure to enable more accurate comparisons. Thirdly, the inclusion and exclusion criteria for the study should be clearly stated in the materials and methods section.

Moreover, while the study identifies notable differences in metabolic parameters between axonal and demyelinating groups, the underlying mechanisms remain unclear. Further investigation is warranted to elucidate the specific biochemical pathways involved in N2O-induced neuropathy and their role in the pathophysiology.

In conclusion, despite providing valuable insights into the metabolic pathophysiology of N2O-induced neuropathy, this study has limitations that need to be addressed in future research. These findings contribute to a better understanding of the biochemical and electrophysiological alterations associated with this condition and may guide future therapeutic interventions. However, larger controlled studies and a more in-depth exploration of the underlying mechanisms are crucial for a more comprehensive understanding.

Author Response

Thank you for your insightful and comprehensive review of our study on the metabolic pathophysiology of nitrous oxide (N2O)-induced neuropathy. Your acknowledgment of the study's contribution to understanding the neurological complications associated with N2O abuse is greatly appreciated. We are grateful for your positive remarks regarding our approach combining metabolic and electrophysiological analyses, and the structured data collection from our defined patient cohort.

In addressing the limitations you've highlighted, we wish to emphasize the challenge of recruiting a larger sample size, given the nature of our study population. The number of subjects included directly correlates with the incidence of patients presenting with N2O-induced neuropathy, where electrophysiological testing is not routinely performed. This limitation impacts our ability to expand the cohort size and complicates the inclusion of a control group without N2O exposure. The specificity of our research criteria, focusing on patients with both confirmed exposure to N2O and subsequent electrophysiological assessment, inherently limits the pool of eligible study participants.

Moreover, the absence of routine electrophysiological testing in these patients adds an additional layer of complexity to gathering concomitant data, which is critical for our analysis. This has constrained our capacity to conduct a larger, controlled study with a clearly defined control group.

We acknowledge your suggestion for a clearer exposition of the inclusion and exclusion criteria and will strive to articulate these more precisely in our manuscript. Regarding the call for further exploration into the biochemical pathways involved in N2O-induced neuropathy, we fully agree and aim to pursue this in future research, which will hopefully illuminate the mechanisms at play more clearly.

Hence, we have added a paragraph at the end of the manuscript which raises these points :

"This study provides preliminary data that underscore the necessity for further confirmation through research on a larger patient cohort. Additionally, our findings suggest the potential need for exploring the neurotoxic effects of homocysteine. The limited scale of our current investigation highlights the importance of expanding our research scope to not only validate these initial observations but also to delve deeper into understanding the intricate relationship between elevated homocysteine levels and its possible neurotoxic implications. Future studies with expanded cohorts are crucial for establishing definitive conclusions and enhancing our comprehension of homocysteine role in nitrous oxide abuse."